# Influence of Left Ventricular Diastolic Dysfunction on the Diagnostic Performance of Coronary Computed Tomography Angiography-Derived Fractional Flow Reserve

**DOI:** 10.3390/jcm12051724

**Published:** 2023-02-21

**Authors:** Zhixin Xie, Tianlong Wu, Jing Mu, Ping Zhang, Xuan Wang, Tao Liang, Yihan Weng, Jianfang Luo, Huimin Yu

**Affiliations:** 1The Second School of Clinical Medicine, Southern Medical University, Guangzhou 510515, China; 2Guangdong Provincial People’s Hospital, Guangdong Academy of Medical Sciences, Guangzhou 510080, China; 3Guangdong Cardiovascular Institute, Guangzhou 510080, China; 4Department of Cardiology, Shenzhen Hospital, Southern Medical University, Shenzhen 518000, China; 5School of Medicine, South China University of Technology, Guangzhou 510006, China; 6Shantou University Medical College, Shantou 515041, China; 7Department of Cardiology, Guangdong Provincial People’s Hospital’s Nanhai Hospital, Foshan 528000, China

**Keywords:** fractional flow reserve, coronary computed tomography angiography-derived fractional flow reserve, left ventricular diastolic dysfunction

## Abstract

Objectives: Our study aimed to demonstrate the influence of left ventricular (LV) diastolic dysfunction on the diagnostic performance of coronary computed tomography angiography-derived fractional flow reserve (CT-FFR). Methods: One hundred vessels from 90 patients were retrospectively analyzed. All patients underwent echocardiography, coronary computed tomography angiography (CCTA), CT-FFR, invasive coronary angiography (ICA), and fractional flow reserve (FFR). The study population was divided into normal and dysfunction groups according to the LV diastolic function, and the diagnostic performance in both groups was assessed. Results: There was a good correlation between CT-FFR and FFR (R = 0.768 *p* < 0.001) on a per-vessel basis. The sensitivity, specificity, and accuracy were 82.3%, 81.8%, and 82%, respectively. The sensitivity, specificity, and accuracy were 84.6%, 88.5%, and 87.2% in the normal group and 81%, 77.5%, and 78.7% in the dysfunction group, respectively. CT-FFR showed no statistically significant difference in the AUC in the normal group vs. the dysfunction group (AUC: 0.920 [95% CI 0.787–0.983] vs. 0.871 [95% CI 0.761–0.943], Z = 0.772 *p* = 0.440). However, there was still a good correlation between CT-FFR and FFR in the normal group (R = 0.767, *p* < 0.001) and dysfunction group (R = 0.767 *p* < 0.001). Conclusions: LV diastolic dysfunction had no effect on the diagnostic accuracy of CT-FFR. CT-FFR has good diagnostic performance in both LV diastolic dysfunction and the normal group and can be used as an effective tool for finding lesion-specific ischemia while screening for arterial disease in patients.

## 1. Introduction

Fractional flow reserve (FFR) is the gold standard for evaluating ischemic lesions [1]. Compared with angiography-guided revascularization strategies, FFR-guided PCI has been shown to improve clinical outcomes with long-term follow-up [2,3]. However, the measurement of FFR is a costly and invasive procedure and can inherently increase the risk of serious complications. In recent years, CT-FFR has been rapidly developing. Several prospective multicenter studies have shown that CT-FFR can be used to identify ischemic stenosis. Compared with FFR, CT-FFR has good diagnostic performance, with high sensitivity and specificity to identify whether there are hemodynamically related disorders, and CT-FFR does not require additional interventional procedures or the use of drugs such as adenosine to induce coronary congestion [4,5,6]. Multiple factors, such as motion artifacts, image quality, calcification, artifacts, and microcirculation disturbance, affect the accuracy of CT-FFR [7,8,9]. The impact of diastolic dysfunction on its diagnostic performance has not been reported thus far.

Studies have shown that microcirculation disturbance is an important mechanism of LV diastolic dysfunction, which is affected by coronary blood congestion [9,10]. The abnormal early diastolic function of patients with coronary heart disease is related to myocardial ischemia and hypoxia caused by coronary stenosis and gradually increases with the expansion of the stenosis range and the aggravation of the degree of stenosis. In addition, diastolic function precedes systolic function. Abnormality occurs [11], so LV diastolic dysfunction may have an impact on CT-FFR values, which, in turn, leads to differences in the diagnostic performance of CT-FFR. Our study aimed to investigate the diagnostic performance of CT-FFR and explore its consistency in LV normal and dysfunctional diastolic function using invasive FFR as the standard of reference.

## 2. Materials and Methods

We conducted a retrospective observational study. The flowchart of the study is presented in Figure 1. The study was approved by the Guangdong Provincial People’s Hospital Ethics Committee. Since the retrospective nature of the study involved only chart review, the patient consent requirement was waived.

### 2.1. Study Population

We retrospectively analyzed 100 vessels from 90 patients who underwent CCTA and coronary angiography (CAG) in Guangdong Provincial People’s Hospital from 2015 to March 2022. These patients underwent CCTA and echocardiography testing 90 days prior to coronary angiography and invasive FFR testing for at least one coronary lesion. The exclusion criteria included: previous coronary artery bypass surgery or stenting, poor echocardiographic images for evaluation, severe valvular disease, severe tortuous or calcified lesions, severe left main disease, severe artifacts, dislocation, noise and calcification present in the CCTA image, causing the CT-FFR analysis to be impossible, and an unqualified pressure curve for FFR analysis.

### 2.2. Coronary Computed Tomography Angiography Analysis

The CTA examination was performed on the patients using Somatom Definition Flash, a Siemens second-generation dual-source CT scanner. Due to the high time resolution of dual-source CT, CCTA has a wide range of applications and does not require human control of the heart rate. Prior to the examination, the patients were trained to hold their breath and were connected to ECG gating. The subject would lay on the examination bed with their head advanced, hands raised, and heart placed in the scanning center. The adaptive prospective ECG gating sequence scanning mode was adopted, relative value scanning was performed, and the scanning time window was set to 30~70%. The scanning range was 1 cm below the tracheal bifurcation to the heart diaphragm. The scanning parameters were as follows: automatic real-time dynamic radiation dose adjustment technology (CAREDose 4D), reference tube voltage 120 kV, reference tube current 380 mAs, elastic rotation time (automatically adjusted according to the heart rate), and width of the collimator 128 × 0.6 mm. The reconstruction parameters were as follows: reconstruction layer thickness 1 mm. The interval between the reconstruction layers was 0.7 mm. The reconstruction algorithm SAFIRE iterative reconstruction was level 3. The contrast agent injection scheme was as follows: 70 mL of the nonionic contrast agent Iopamiro 370 was injected through a cubital vein at a speed of 5 mL/s. After the contrast agent injection, 30 mL normal saline was used to flush the tube at a flow rate of 5 mL/s. Blous tracking technology was used to start scanning. The contrast agent was injected, and tracking started 10 s later. The tracking level was 1 cm below the tracheal bifurcation, and the area of interest was located in the ascending aorta. There were four groups of images of the automatic optimal diastolic period, automatic optimal systolic period, and manual 45% and manual 75% were reconstructed after scanning. All images were uploaded to the workstation, and the images were reconstructed and analyzed by 2 experienced radiologists.

### 2.3. CT-FFR Analysis

The CT-FFR calculation and coronary artery remodeling were performed by Ray-sight Inc. using a blinded protocol. The process involved four main steps: (1) anatomic model reconstruction, (2) centerline definition, (3) boundary condition, and (4) CT-FFR calculation.

Anatomic model reconstruction. The 3-dimensional patient-specific model, including the heart and coronary artery tree, was automatically derived from CTA. During the model construction, the Frangi algorithm was used to extract the chambers, myocardium, and aorta. Coronary artery segmentation was performed using an automatically generated centerline model and grown. For the finite element method used in calculating CT-FFR, a mesh model including millions of vertices was generated from the geometric model of the aorta and coronary artery tree.

Centerline definition. On the basis of the anatomic model, we extracted the centerline of each coronary artery, which was useful for locating the boundary regions (coronary outlets) and setting boundary conditions. First, the cross-sectional image perpendicular to the centerline was reconstructed, and a region-of-interest contour (1 mm^2^) located at the center of the cross-sectional image was defined. Then, the mean Hounsfield unit value of the region of interest was calculated from the ostium to the distal level, where the vessel cross-sectional area fell below 2.0 mm^2^.

Boundary conditions. We used patient-individualized brachial pressure to derive the aortic pressure, and the mean aortic pressure was coupled at the inlet boundary. As for the outlet boundary conditions, the blood flow based on the cardiac output and myocardial mass were computed and coupled. Specifically, the total resting blood flow (including aorta and coronary) was estimated from the cardiac output, while the total resting coronary blood flow was computed using the myocardial mass. To estimate hyperemia, the distal resistance of each artery and aortic pressure were reduced to 0.24 times and 0.8 times the resting state value, respectively [12].

CT-FFR calculation. A CFD simulation was performed on a standard desktop workstation that used a finite volume approach to solve the Navier–Stokes equations. Blood was treated as a noncompressible, viscous Newtonian fluid, and the vessel wall was assumed to be rigid, with a no-slip boundary condition. Finally, the velocity and pressure at each vertex of the mesh model was generated, and the CT-FFR values were calculated as the pressure ratio. The CT-FFR software used in this study has been validated previously in a prospective, multicenter study [13].

### 2.4. Invasive Angiography and FFR Measurement

Invasive angiography was recorded by a digital subtraction angiography machine (Allura, Philips, Amsterdam, the Netherlands) at 15 frames, either through the femoral or the radial approach. Nitroglycerin was injected into the coronary artery before coronary angiography, and manual or high-pressure injection via a syringe was performed for injecting the nonionic contrast medium. In this study, FFR (RadiAnalyzer Xpress) was performed during ICA in at least 1 vessel with a diameter 2 mm and 10% to 90% visual stenosis and was chosen at the discretion of the operator blinded to the CT findings. The pressure wire (St. Jude’s Medical) was calibrated and electronically equalized with the aortic pressure before being placed in the distal third of the coronary artery being interrogated. Intracoronary glyceryl trinitrate (100 mm) was injected to minimize vasospasm. Intravenous adenosine was administered (140 mm/kg per min) through an intravenous line in the antecubital fossa. At steady-state hyperemia, FFR was recorded and calculated by dividing the mean coronary pressure measured with the pressure sensor placed distally to the stenosis by the mean aortic pressure measured through the guide catheter. The pressure sensor was then pulled back into the tip of the guiding catheter, and only runs with ≤0.03 drift were accepted for analysis.

### 2.5. Echocardiographic Assessment of Left Ventricular Diastolic Function

The Philips i E33 color Doppler ultrasound system, probe S5-1, frequency 1~5 MHz and the GE vivid dimension color Doppler ultrasound system, probe M3S, frequency 1.5~4.2 MHz were used in this study. According to the 2016 American Echocardiography Society (ASE) guidelines [14], the assessment of diastolic dysfunction included an evaluation of the septal/lateral tissue Doppler imaging (TDI) e0 velocity, E/e0 ratio, mitral valve E/A ratio, tricuspid regurgitation (TR) velocity, and left atrial volume index (LAVI).

### 2.6. Statistical Analysis

Continuous variables were expressed as the mean ± standard deviation or the median (interquartile range), categorical variables were expressed as frequencies (percentages), and the Kolmogorov–Smirnov test was used to assess the normality of the distribution of the continuous variables. Normally or nonnormally distributed variables were compared using independent samples *t*-tests or Mann–Whitney tests, respectively. FFR was indicated as the gold standard diagnosis of lesion-specific ischemia with a cutoff value of 0.80, which was consistent with most contemporary studies. The diagnostic performance of CT-FFR is expressed by a receiver operating characteristic curve, which is acceptable if the area under the curve (AUC) is greater than 0.70. Spearman’s correlation was used to analyze the correlation between CT-FFR and invasive FFR values. The Bland–Altman statistic was used to plot the difference between CT-FFR and mean invasive FFR. The Z test was performed to compare the diagnostic performance of CT-FFR in the patients in the normal and dysfunction groups. All statistics were performed using IBM SPSS version 25 and MedCalc software.

## 3. Results

### 3.1. Baseline Characteristics

The patient characteristics and echocardiographic parameters are shown in Table 1. A total of 67.8% of the patients were male, the mean age was 64.1 years, 70 vessels were LAD, 12 vessels were LCX, 18 vessels were RCA, and the patients with normal diastolic function were assigned to one group (N = 39). The patients with dysfunctional diastolic function were assigned to the second group (n = 61). Figure 2 shows typical cases examined by CCTA, CT-FFR, and CAG in the left anterior descending branch.

### 3.2. Diagnostic Performance of CT-FFR Compared with FFR

Consistent with most contemporary studies, we used an FFR cutoff of 0.80, and the following analysis was performed at the per-vessel level. Table 2 shows CT-FFR in relation to FFR, and the sensitivity, specificity, accuracy, positive predictive value, and negative predictive value were 82.3%, 81.8%, 82%, 70%, and 90%, respectively. As shown in Figure 3, the CT-FFR was well correlated with the invasive FFR (R2 = 0.768, *p* < 0.001), and the AUC was 0.892 [95% CI 0.814–0.945]. Further analysis of the systematic differences was performed, which indicated that the mean difference between FFR and CT-FFR was 0.036, and the 95% confidence interval was 0.022 to 0.05.

### 3.3. Diagnostic Performance and Correlation of CT-FFR to Invasive FFR between the Normal and Dysfunction Groups in Left Ventricular Diastolic Function

As shown in Table 2, the sensitivity, specificity, accuracy positive predictive value, and negative predictive value were 84.6%, 88.5%, 87.2%, 78.6%, and 92% in the LV diastolic normal group vs. 81%, 77.5%, 78.7%, 65.4%, and 88.6% in the dysfunction group. As shown in Figure 4, CT-FFR showed no statistically significant difference in the area under the receiver operating characteristic curve (AUC) in the normal group vs. the dysfunction group (AUC: 0.920 [95% CI 0.787–0.983] vs. 0.871 [95% CI 0.761–0.943], Z = 0.772, *p* = 0.440). There was still a good correlation between CT-FFR in the normal group (R = 0.767, *p* < 0.001), mean difference: 0.035, and 95% confidence interval 0.018–0.053 vs. the dysfunction group (R = 0.767, *p* < 0.001), mean difference: 0.037, and 95% confidence interval 0.017–0.057.

## 4. Discussion

Our study focused on the diagnostic performance of CT-FFR and found that CT-FFR had good diagnostic accuracy compared with FFR, which was maintained in the subgroup analysis of different diastolic functions. LV diastolic dysfunction did not affect the diagnostic performance of CT-FFR.

FFR is currently the “gold standard” for assessing whether coronary stenosis leads to myocardial ischemia. Under the guidance of FFR, the effectiveness of a coronary intervention can be increased, and unnecessary stenting can be reduced. However, FFR is an invasive technique that requires a pressure guide wire to be placed into the diseased vessel through a catheter under coronary angiography. The cost is expensive, and there is a risk of damage to the vessel during the operation, which limits the clinical application of this test in China [15,16,17]. In recent years, coronary computed tomography angiography-derived fractional flow reverse (CT-FFR) has injected new vitality into the assessment of coronary function. Different from the traditional and invasive gold standard for coronary functional assessment, CT-FFR technology is a noninvasive detection method based on coronary CT that combines coronary anatomy and a functional assessment. The functional analysis of simulated fluid dynamics has the advantages of CTA and FFR at the same time. It has certain advantages in identifying vascular stenosis and guiding treatment strategies. Early DISCOVER-FLOW and DeFACTO studies confirmed that CT-FFR has good diagnostic efficiency, sensitivity, and specificity [4,18]. CT-FFR is mainly used in outpatient clinics and has been shown to reduce the number of unnecessary CAG procedures in patients with nonfunctional significant CAD [19,20]. Donnelly et al. [21] demonstrated CT-FFR with 91% sensitivity, 72% specificity, and 78% accuracy; van Hamersvelt et al. [22] also demonstrated that CT-FFR has a sensitivity of 89%, a specificity of 78%, and an accuracy of 83%, all comparable to the results of our current study. We also observed a good correlation between CT-FFR and FFR by simple linear analysis and Bland–Altman plots. This is consistent with our research.

Studies have shown that microcirculation disturbance is an important mechanism of LV dysfunction [9,10]. Long-term microcirculation disturbance will cause myocardial ischemia and hypoxia, which will cause myocardial compensatory hypertrophy, and eventually, decompensation of the cardiac function will occur with the progression of the disease. Due to the imbalance of myocardial blood supply and oxygen supply and abnormal ventricular wall motion, abnormal calcium ion transport in myocardial cells and poor diastolic coordination affect the cardiac compliance, resulting in cardiac diastolic and systolic dysfunction. A number of previous studies have shown that, in the early stage of coronary heart disease, the diastolic function is abnormal before the systolic function [23]. Therefore, the abnormal diastolic function in patients with coronary heart disease in the early stage is more related to the myocardial ischemia and hypoxia caused by coronary stenosis. Ryberg et al. believed that LV diastolic function is closely related to the degree of coronary stenosis [24], and it gradually increases with the expansion of the stenosis and the aggravation of the degree of stenosis. Bogsert et al. concluded that premature intervention causing diastolic dysfunction in stenotic vessels rescues ischemic cardiomyopathy [11]. Ren et al. found a strong association of moderate to severe left ventricular diastolic dysfunction with heart failure hospitalization events and cardiac death [25]. For patients with LV diastolic dysfunction who are suspected of having coronary heart disease, timely diagnosis and treatment, such as revascularization, are closely related to the prognosis of the patients [26]. Williams and Kim successively confirmed the presence of localized diastolic dysfunction in patients with coronary heart disease using different methods of echocardiography, and reduced diastolic function can be used as a sensitive indicator of myocardial ischemia [27,28].

Hemodynamic factors, induced by pulsatile blood flow, play a crucial role in vascular health and diseases, such as the initiation and progression of atherosclerosis. Computational fluid dynamics, finite element analysis, and fluid–structure interaction simulations have been widely used to quantify detailed hemodynamic forces based on vascular images commonly obtained from computed tomography angiography, magnetic resonance imaging, ultrasound, and optical coherence. Its steps include medical imaging, image processing, spatial discretization to generate computational mesh, setting up boundary conditions and solver parameters, visualization and extraction of hemodynamic factors, and statistical analysis [29,30]. Based on coronary computed tomography angiography-derived fractional flow reserve (CT-FFR) combined with coronary CTA anatomy and FFR functional evaluation, no special scanning protocol and the use of additional drugs are required, only based on resting CTA data, computational fluid dynamics (CFD) method is used to simulate intracoronary blood flow and pressure, and then after complex image processing and operation processes (including image segmentation and extraction of the coronary tree, maximum blood flow estimation, computational fluid dynamics evaluation, etc.), FFR at any point of the coronary tree can be obtained, the values are affected by various factors, such as motion artifacts, image quality, calcification artifacts, microcirculatory disturbances, etc. In patients with microvascular disease, the adenosine-mediated hyperemia model may overestimate the degree of vasodilation, resulting in lower CT-FFR values than the measured FFR values [7,8,31], so LV diastolic dysfunction may have an impact on the CT-FFR values, leading to differences in the diagnostic performance of CT-FFR. Hassan Tahir et al. [32] found that diastolic dysfunction is an important risk factor leading to discordance between iFR and FFR. Kawata et al. also found that coronary flow reserve (CFR) is associated with left ventricular diastolic dysfunction in patients with type 2 diabetes [33]. The accuracy of CT-FFR has not been reported in patients with normal vs. impaired left ventricular diastolic function. Our study is the first to analyze the diagnostic performance of CT-FFR in different LV diastolic functions, and we divided the patients into a normal LV diastolic function group and a dysfunction group according to echocardiography. Statistical analysis was performed on the two groups; the results showed that CT-FFR and FFR had good consistency and diagnostic performance in the two groups; and there was no significant difference between them.

Limitations of this study: First, this study is a retrospective study including all patients undergoing echocardiography, CCTA, CT-FFR, ICA and FFR, 46 out of 136 patients were excluded because they had not undergone echocardiography, had a history of PCI, and failed the CT-FFR analysis; thus, potential selection bias can affect the diagnostic accuracy of CT-FFR in identifying lesion-specific ischemia. Second, the sample size was relatively small, and thus, the results still need to be confirmed in a larger sample-sized study. Third, CT-FFR accuracy relies more on high-quality CTA images, and microcirculatory disturbances affect them less. Finally, the results still need to be confirmed by studies with larger sample sizes.

## 5. Conclusions

LV diastolic dysfunction did not affect the diagnostic performance of CT-FFR. CT-FFR has good diagnostic performance in both LV diastolic dysfunction and normal LV, and CT-FFR can be used as an effective tool for screening disease-specific ischemia in patients with coronary artery disease.

## Figures and Tables

**Figure 1 jcm-12-01724-f001:**
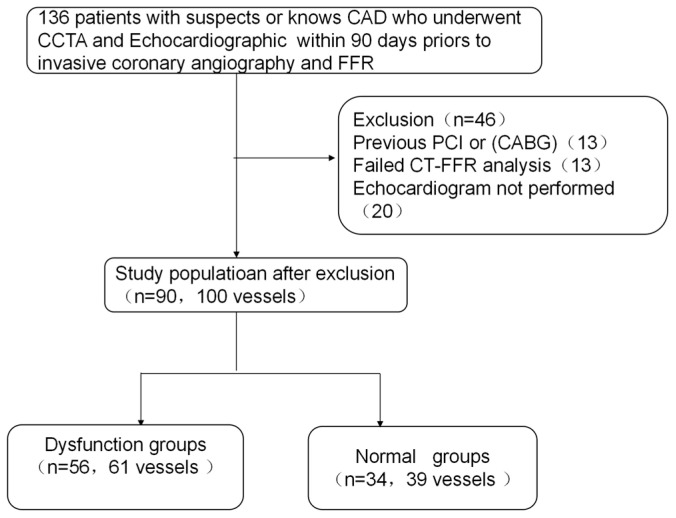
The flowchart of this study. Abbreviations: CAD, coronary heart disease; CABG, coronary artery bypass graft.

**Figure 2 jcm-12-01724-f002:**
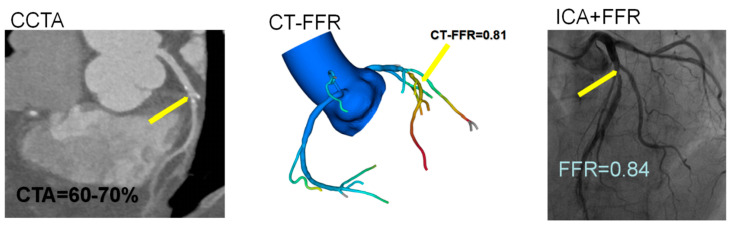
Typical case in a 67-year-old woman showing moderate (60–70%) stenosis in the proximal LAD on CCTA. The CT-FFR and invasive FFR measured distal to the lesion were 0.81 and 0.84, respectively.

**Figure 3 jcm-12-01724-f003:**
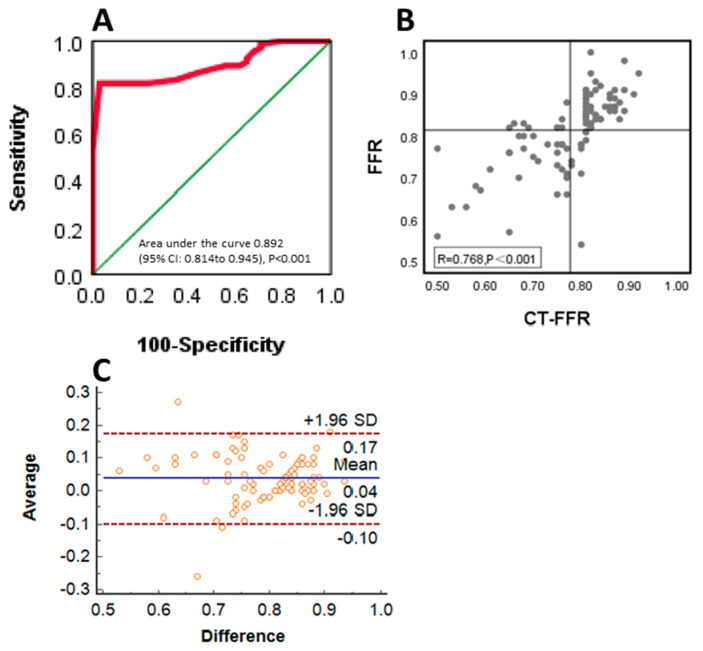
Receiver operating characteristic (ROC) curve (**A**), linear correlation plot (**B**), and Bland–Altman plot (**C**) of resting CT-FFR vs. FFR on a per-vessel basis.

**Figure 4 jcm-12-01724-f004:**
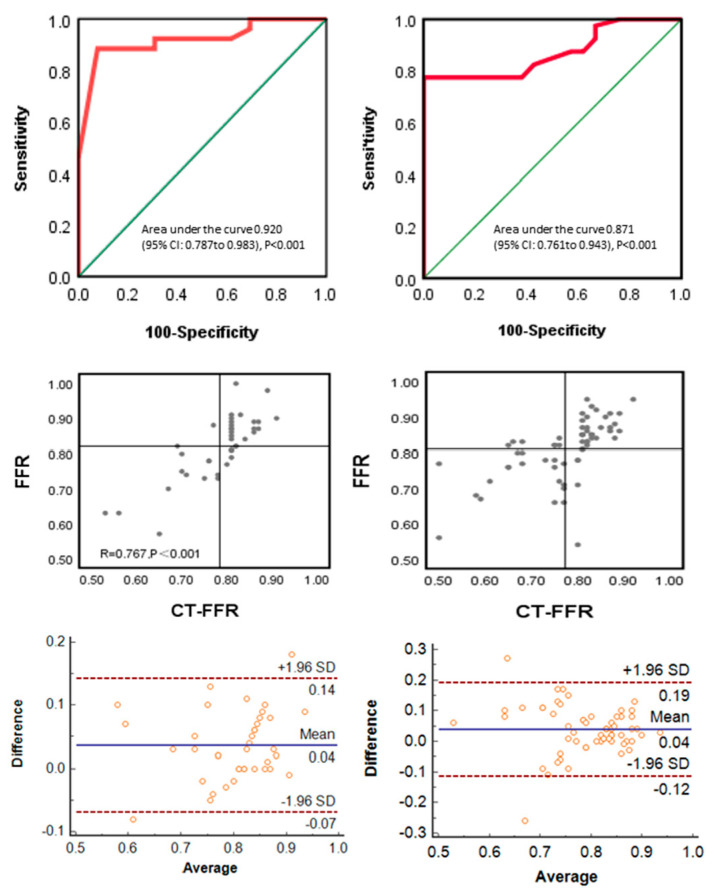
Receiver operating characteristic curve (ROC), linear correlation plot, and Bland–Altman plot comparing CT-FFR vs. FFR on a per-vessel basis in the (**left-hand**) normal and (**right-hand**) dysfunction subgroups in left ventricular diastolic function separately.

**Table 1 jcm-12-01724-t001:** Baseline clinical characteristics.

Basic Characteristics	All Patients	Dysfunction Group	Normal Group	*p* Value
No. of patients	90	56	34	-
No. of vessels	100	39	61	-
Age (years)	64.1 ± 9.64	66 ± 10.6	60.94 ± 6.6	0.006
Man, n (%)	61 (67.8)	36 (64.3)	25 (73.5)	0.363
BMI (kg/m^2^)	23.6 (22.3–25.3)	23.7 (22.8–25.7)	23.5 (22–24.7)	0.296
TC (mmol/L)	4.1 (3.5–4.8)	4.1 (3.5–4.5)	4.1 (3.4–5.2)	0.484
TG (mmol/L)	1.23 (0.84–1.83)	1.23 (0.85–1.88)	1.28 (0.83–1.73)	0.970
LDL-C (mmol/L)	2.53 (2.1–3.19)	2.52 (2.16–2.86)	2.53 (2.02–3.55)	0.671
HDL-C (mmol/L)	1.01 (0.89–1.21)	1.00 (0.91–1.17)	1.06 (0.86–1.24)	0.758
Cr (umol/L)	76.2 (64.1–90.8)	78.7 (67–94.1)	73.4 (61.3–87.2)	0.193
Pertinent medical history, n (%)				
Hypertension	52 (57.8)	38 (67.9)	14 (41.2)	0.013
Hyperlipidemia	32 (35.6)	23 (41.1)	9 (26)	0.161
Family history of CAD	2 (2.2)	1 (1.8)	1 (2.9)	0.404
smoker	18 (20)	8 (14.3)	10 (29.4)	0.082
Diabetes	16 (17.8)	12 (21.4)	4 (11.8)	0.380
Cerebral infarction	7 (7.8)	6 (10.7)	1 (2.9)	0.353
Echocardiographic Parameters				
LVEF (%)	64.9 ± 6.1	64.2 ± 6.6	66.2 ± 5.1	0.064
E/e′	11.8 ± 3.9	13.3 ± 4.04	9.1 ± 1.6	0.001
E/A	0.85 ± 0.29	0.85 ± 0.3	0.86 ± 0.2	0.9
e′	6.0 ± 1.5	5.4 ± 1.1	7.1 ± 1.59	0.001
IVST (mm)	10.0 ± 1.1	10.2 ± 1.1	9.65 ± 1.08	0.038
LVEDD (mm)	45.7 ± 5.4	45.5 ± 5.8	46.2 ± 7.1	0.274
LVESD (mm)	28.4 ± 5.1	28.4 ± 5.9	28.5 ± 3.5	0.484
LVPW (mm)	9.8 ± 1.2	10 ± 1.1	9.5 ± 1.2	0.046
Vessel location, n (%)				
LAD	70 (70)	44 (72.1)	26 (66.7)	0.816
LCX	12 (12)	8 (13.1)	4 (10.2)	0.983
RCA	18 (18)	9 (14.8)	9 (23.1)	0.232
FFR	0.81 ± 0.87	0.81 ± 0.85	0.82 ± 0.90	0.586
CT-FFR	0.77 ± 0.88	0.77 ± 0.92	0.78 ± 0.81	0.533

Abbreviations: BMI, Body mass index; TC, Total cholesterol; TG: Triglyceride; LDL-C, Low-density lipoprotein cholesterol; HDL-C, High-density lipoprotein cholesterol; Cr, Creatinine; CAD, Coronary artery disease; LVEF, Left ventricular ejection fraction; LVEDD, Left ventricular end-diastolic dimension; LVESD, Left ventricular end-systolic dimension; LVPW: Left ventricular posterior wall; IVST, Interventricular septal thickness; LAD, Left anterior descending artery;LCX, Left circumflex artery; RCA, Right coronary artery; CT-FFR, Coronary CT angiography–derived fractional flow reserve; FFR, Fractional flow reserve.

**Table 2 jcm-12-01724-t002:** Diagnostic performance of CT-FFR for the detection of lesion-specific ischemia on a per-vessel level with invasive FFR as a reference standard.

Analysis Basis		Results										
		Modality	TP	TN	FP	FN	Sen.	Spec.	Acc.	PPV	NPV	AUC
Total	Per-vessel	CT-FFR	28	54	12	6	82.3	81.8	82	70	90	0.89
normal group	Per-vessel	CT-FFR	11	23	3	2	84.6	88.5	87.2	78.6	92	0.920
dysfunction group	Per-vessel	CT-FFR	17	31	9	4	81	77.5	78.7	65.4	88.6	0.871

Abbreviations: TP, true positive; TN, true negative; FP, false-positive; FN, false-negative; Sen., sensitivity; Spec., specificity; Acc., accuracy; PPV, positive predictive value; NPV, negative predictive value; AUC, area under the curve.

## Data Availability

The data presented in this study are available on request from the corresponding author. The data are not publicly available due to privacy issues.

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
