# Peer review of "Influence of Left Ventricular Diastolic Dysfunction on the Diagnostic Performance of Coronary Computed Tomography Angiography-Derived Fractional Flow Reserve"

_jcm, 2023, doi:10.3390/jcm12051724_

Round 1

Reviewer 1 Report

Dear Authors,

in your manuscript, you evaluate the influence of diastolic dysfunction on the diagnostic accuracy of CT-FFR, what is an interesting topic.

There are several things to consider. For better read- and understandability please explain every abbreviation before first usage, please extinguish the (many) typos and grammatical deficiencies.

Major:

- you mention, that the use of drugs (adenosine) is a disadvantage of FFR. you do this without mentioning RFR, iFR. you mention that multiple factors contribute to accuracy (lines 49f): how do they contribute?

- 46 of 136 patients were excluded. you need to include this in the limitations section.

- Coro FFR: you really injected 140µg/kg per minute i.c.?

- diastolic dysfunction: please state the parameters somewhere in the script (e.g. baseline); an original tracing would be fine in this regard. is there a difference in severity of diastolic dysfunction?

Minor:

- exactly 90 days? (line 73)

- figures are hard to read, please improve

Author Response

Dear Reviewers,Thanks very mush for taking your time to review this manuscript.I really appreciate all your comments and suggestions!Please find my itemized responses in below and my revisions in the re-submitted files.

Response 1:we regret there were problems with the English.The paper has been carefully revised by a professional language editing service to improve the grammar

Response 2:Thank you this valuable feedback,FFR is currently the "gold standard" for assessing whether coronary stenosis leads to myocardial ischemia.The correlation between iFR and ffr has been previously demonstrated(Tahir H,et al. (2021) Association of Echocardiographic Diastolic Dysfunction with Discordance of Invasive Intracoronary Pressure Indices. J Clin Med 16),Our study aimed to demonstrate the influence of left ventricular (LV) diastolic dysfunction on the diagnostic performance of CT-FFR.CT-FFR calculation process is to use artificial intelligence and image processing technology to extract 3D anatomical model of coronary artery from CT, perform physiological modeling for the relationship between aorta, blood vessel and microvessel, apply machine learning and hydrodynamic technology to calculate the flow velocity pressure of blood in blood vessel, therefore, motion artifacts, image quality, calcification, artifacts, and microcirculation disturbancet will affect the CT-FFR calculation process.

Response 3:  Thank you for the suggestion,we have modified this expression in the limitations section.

Response 4: Yes,invasive FFR were performed according to standard practice,The strength of cardiology department of Guangdong Provincial People's Hospital is the top five in China, and our operations are strictly performed by experienced doctors.

Response 5: Thank you for the suggestion,We have added left ventricular diastolic dysfunction parameter in baseline.

Response 6:  Yes,all patients underwent CCTA and echocardiography testing 90 days prior to coronary angiography. and invasive FFR testing for at least one coronary lesion.patients who were excluded 90 days later.

Response 7:Thank you this valuable feedbackhank,The quality of the figures and their description has been carefully revised.

Reviewer 2 Report

Interesting investigation on the influence of left ventricular diastolic dysfunction on the diagnostic performance of coronary fractional flow reverse. The aim of the paper is clear and interesting to me, but the paper can be considered for publication after the authors have replied to the following mandatory remarks:

1) Moderate English changes required to improve the manuscript

2) The setup should be better described. The authors state: “Navier-Stokes equations were solved by a finite element algorithm assuming the blood flow as an incompressible Newtonian fluid with certain viscosity at each point in the 3D geometric model, and the corresponding pressure and flow velocity were thus acquired”. Which kind of numerical approach do they use to solve the NS equations? Space and time discretization? Boundary conditions? Grid independency analysis? 

3) The authors should provide a better description of the methodological procedure, and, in particular, of the synergic use of imaging and simulations. Many studies in the literature provide this integration. Is their procedure like the ones presented in A. Boccadifuoco et al. (2016), A. Mariotti et al. (2021) and Y. He et al. (2022)? The authors should discuss this point and include the suggested references and possibly others. 

4) The quality of the figures and their description should be improved. 

References

Y. He et al. (2022) "Medical image-based computational fluid dynamics and fluid-structure interaction analysis in vascular diseases". Front. Bioeng. Biotech. 671.

A. Mariotti et al. (2021) "Hemodynamics and stresses in numerical simulations of the thoracic aorta: Stochastic sensitivity analysis to inlet flow-rate waveform". Comp. Fluids, 230, 105123.

A. Boccadifuoco et al. (2016) “Uncertainty quantification in numerical simulations of the flow in thoracic aortic aneurysms”. ECCOMAS Congress 2016 - Proceedings of the 7th European Congress on Computational Methods in Applied Sciences and Engineering, 2016, 3, pp. 6226–6249

Author Response

Dear Reviewers,Thanks very mush for taking your time to review this manuscript.I really appreciate all your comments and suggestions!Please find my itemized responses in below and my revisions in the re-submitted files.

Response 1:we regret there were problems with the English.The paper has been carefully revised by a professional language editing service to improve the grammar.

Response 2:Thank you this valuable feedback,1、Finite Element Method 2、Space was discretized into tetrahedrons. Time was discretized using an implicit fashion. For more details, please refer:E. Jansen, “A stabilized finite element method for computing turbulence,” Computer Methods in Applied Mechanics and Engineering, vol. 174, no. 3, pp. 299–317, May 1999   3、Inlet boundary conditions: patient-specific flow rate, fixed pressure; Outlet boundary conditions: Windkessel 4、Grid independency was performed. The chosen mesh size had only 0.2% error compared with the finest mesh. Note that local mesh refinement was applied in stenosis regions.

Response 3 We have consulted a large body of literature on methodology and revised the section on methodological procedures, since the main purpose of our study was to demonstrate the influence of left ventricular (LV) diastolic dysfunction on the diagnostic perfor-mance of CT-FFR, it was not elaborated discuss the methodological procedure.

Response 4:Thank you this valuable feedbackhank,The quality of the figures and their description has been carefully revised.

Reviewer 3 Report

Authors present that coronary CT-FFR correlates well with invasive FFR, which is not influenced by the diastolic dysfunction. Data are clear.

They should highlight that these observations are not studied in patients with reduced ejection fraction, where diastolic dysfunction might have a much higher impact on FFR, theoretically.

Author Response

Dear Reviewers,Thanks very mush for taking your time to review this manuscript.I really appreciate all your comments and suggestions!Please find my itemized responses in below and my revisions in the re-submitted files.

Response 1:Thank you this valuable feedback,Our study included patients with reduced ejection fraction because there are currently numerous studies on FFR that do not exclude patients with reduced ejection fraction, and the impact of left ventricular ejection fraction on fractional flow reserve: Insights from the FAME (Fractional Flow versus Reserve Angiography for Multivessel Evaluation) trial concluded that Reduced EF has no value on the FFR value.

Round 2

Reviewer 2 Report

The authors do not respond to point (3) and they do not include the answer to point (2) in the revised manuscript. The authors do not show interest in the methodological procedure which is a fundamental aspect of a rigorous research paper. The authors should include the answers to mandatory remarks 2 and 3 in the revised manuscript and the suggested references. 

Author Response

Thank you for your valuable feedback,and in response to your questions,we have optimized the methodological section on the manuscript(starting line 110),and we discuss the synergistic use of imaging and simulation and cite the relevant literature( starting line 289)

Please review my revisions in the resubmitted document.